# The Impact of Norepinephrine on Mono-Species and Dual-Species Staphylococcal Biofilms

**DOI:** 10.3390/microorganisms9040820

**Published:** 2021-04-13

**Authors:** Sergey Vladislavovich Mart’yanov, Ekaterina Alexandrovna Botchkova, Vladimir Konstantinovich Plakunov, Andrei Vladislavovich Gannesen

**Affiliations:** 1Laboratory of Viability of Microorganisms, Federal Research Center “Fundamentals of Biotechnology” of Russian Academy of Sciences, 117312 Moscow, Russia; semartyan@inbox.ru (S.V.M.); plakunov@inmi.ru (V.K.P.); 2Laboratory of Microbiology of Anthropogenic Habitats, Federal Research Center “Fundamentals of Biotechnology” of Russian Academy of Sciences, 117312 Moscow, Russia; botchkovaekat@gmail.com

**Keywords:** biofilms, dual-species biofilms, norepinephrine, hormones, skin microbiota, *Staphylococcus epidermidis*, *Staphylococcus aureus*

## Abstract

The effect of norepinephrine (“NE”) on Gram-negative bacteria is well characterized; however, little is known about the impact of NE on cutaneous Gram-positive skin residents, especially staphylococci. In this study, the impact of NE on monospecies and dual-species biofilms of *Staphylococcus epidermidis* and *S. aureus* model strains was investigated for the first time. Biofilms were grown in two different models (on polytetrafluoroethylene (“PTFE”) cubes and glass microfiber filters (“GMFFs”)) and additionally kinetic measurements of bacterial growth was performed. We have shown that NE can affect the biofilm formation of both species with a strong dependence on aerobic or anaerobic culture conditions in different models. It was shown that *S. epidermidis* suppresses *S. aureus* growth in dual-species biofilms and that NE can accelerate this process, contributing to the competitive behavior of staphylococci.

## 1. Introduction

Staphylococci are a well-known genus of Gram-positive pathogenic and saprotrophic bacteria in the phylum Firmicutes. They are commensal microorganisms of the skin microbiota that form multi-species biofilms with other cutaneous bacteria and interact with human humoral regulatory systems and hormones, especially catecholamines [1,2]. Staphylococci are characterized as forming cellular clusters and as facultative anaerobic nonmotile microorganisms. Despite the increased danger of infection, staphylococci are also considered to be components of the normal microbiota of skin and mucosal membranes. *Staphylococcus epidermidis* represents a normal skin commensal, as is *Cutibacterium acnes*. *S. epidermidis* can prevent the reproduction of more virulent *Staphylococcus aureus* and *S. epidermidis* strains [3] through different mechanisms, including prevention of biofilm formation and production of bacteriocins. However, under certain conditions, commensal staphylococci switch to pathogenic behavior and cause infections [4].

Biofilms are a basic product of the bacterial lifestyle in nature and, particularly, on skin. Since bacteria encounter and are influenced by compounds secreted by the human body (host stress mediators such as catecholamines and natriuretic peptides (NUPs)), studying the potential impact of these compounds on biofilm growth is of great importance for the medical and the cosmetic industries. Recent studies have highlighted the role of natriuretic peptides [5], and calcitonin gene-related peptide (CGRP) in staphylococcal biofilms [6]. Catecholamines constitute a highly conserved stress-related group of hormones and neurotransmitters. They are also involved in different processes in microbial cells, including biofilm formation [2].

Norepinephrine (NE) is the most studied catecholamine and plays a role as a neurotransmitter and stress hormone. This compound is abundant in the gastrointestinal tract and therefore interacts with microbial cells. A special two-component system of analogs of adrenergic receptors that sense NE, the QseC/QseB sensor kinase system [7,8,9,10], has recently been discovered and characterized for gram-negative enterobacteria, e.g., *Escherichia* and *Salmonella*. However, little is known about NE interactions with Gram-positive bacteria. The impact of catecholamines was described for the first time by Neal and colleagues, who discovered that catecholamines, especially norepinephrine, facilitate access to the iron supply and can induce the growth of staphylococci [11]. Slightly later, *S. epidermidis* biofilm stimulation by catecholamine ionotropes was observed [12].

In this work, we studied the impact of NE on the growth of planktonic cultures and monospecies and dual-species biofilms of *S. epidermidis* and *S. aureus* strains. The main goal was to gain insight into how this stress hormone affects the biofilm growth of these two strains in mono-species and dual-species culture to determine whether this hormone has a regulatory effect on the bacterial community. Both aerobic and anaerobic conditions were used to model various skin regions.

## 2. Materials and Methods

### 2.1. Bacterial Strains and Culture Conditions

*S. aureus* 209P (ATCC 6538P, type strain), and *S. epidermidis* ATCC 14990 (type strain, isolated from nasal mucosa) were stored at room temperature in semisolid lysogeny broth (LB, Dia-M, Moscow, Russia) with 0.5% agar covered with sterile mineral oil. Both of these strains are able to form well-established biofilms which was shown in previous studies [13,14,15]. Cultures were prepared in LB and grown overnight with shaking at 150 rpm. The incubation temperature was 33.5 °C, which is close to the temperature on the skin surface. Reinforced clostridial medium (RCM) was used for all experiments (peptone (Dia-M)—10 g, yeast extract (Dia-M)—13 g, glucose (Dia-M)—5 g, NaCl (Dia-M)—5 g, sodium acetate (Dia-M)—3 g, starch (Dia-M)—1 g, cysteine-HCl (Biomerieux, Marcy l’Etoile, France)—0.5 g per 1 L of distilled water, pH 6.8–7.0.) as in our previous studies [5,13]. Sterilization was performed at 112 °C for 30 min to avoid sugar caramelization.

### 2.2. Norepinephrine

Norepinephrine (NE, Merck, Darmstadt, Germany) solution was prepared in sterile Milli-Q water and stored at −18 °C. Since NE concentration on skin enables proper identification, a concentration of 600 pg/mL 3.5 × 10^−9^ М) was established to be similar to physiological levels in blood plasma, as indicated by previous study [16]. Also three increased concentrations (3.5 × 10^−8^, 3.5 × 10^−7^, 3.5 × 10^−6^ M) were used.

### 2.3. Polytetrafluoroethylene (PTFE) Cube Assay

A PTFE cube assay was used to compare mono-species planktonic and biofilm growth and to analyze the range of NE concentrations on bacteria. It was developed recently [17,18,19] and used in this work with modifications. Briefly, glass tubes were filled with cubes (21 chemically pure 4 × 4 × 4 mm PTFE cubes, Ftoroplast. tech., Saint Petersburg, Russia). Three milliliters of RCM was added for aerobic cultivation, and 21 mL RCM was added for anaerobic cultivation (10 mL before and 11 mL of sterile RCM after sterilization). Then, 350 μL of prepared cell suspension with OD_540_ = 0.5 was added to each tube, and negative controls were placed in tubes without bacterial inoculation. The tubes were incubated at 33.5 °C while rotating at 180 rpm for 24 or 72 h. Then, the OD_540_ of the planktonic culture was measured against sterile RCM, and the cubes were washed gently twice with tap water to remove residual planktonic culture and then fixed in 96% ethanol for 20 min. After fixation, the cubes were dried and then stained with 0.5% crystal violet (CV) for 15 min. Next, the CV was removed, the cubes were washed 5 times with tap water, and the remaining CV stain in the biofilms was extracted with 3 mL of 96% ethanol, and the absorbance was measured at OD_590_.

### 2.4. Kinetic Measurements of Bacterial Growth in Two Different Models

Cultivation in 96-well plates was set up under both aerobic and anaerobic conditions in two different models. The first model (model I) allowed us to study the dynamic balance of biofilms and planktonic cultures where the planktonic cultures predominated the biofilm on the bottom and side surfaces of the well. Cultures of *S. aureus* and *S. epidermidis* were adjusted to OD_540_ = 0.5 and 1 with sterile physiological saline (PS, 1% NaCl in distilled water). Cultures with OD_540_ = 0.5 were used to study mono-species cultures, and cultures with OD = 1 were mixed in equal proportions to make dual-species communities. Two hundred microliters of RCM was added with and without NE at the appropriate concentration per well, and 3.3 μL of a cell suspension was inoculated.

The second static model (model II) potentially allows us to study the mixture of biofilms and planktonic cultures where the biofilms are predominant. Cultures of *S. aureus* and *S. epidermidis* were washed twice with sterile PS, centrifuged for 15 min at 5000× *g* and 4 °C and adjusted to OD_540_ = 1 and 2 with sterile PS. Cultures with OD_540_ = 1 were used to study mono-species cultures, and cultures with OD_540_ = 2 were mixed in equal proportions to make dual-species communities. Two hundred microliters of the appropriate suspension was inoculated per well for 2 h to allow cells to adhere to the well bottom. Then, suspensions were removed from the wells, the wells were washed with 200 µL of sterile PS, and inoculated with 200 μL of RCM per well with or without NE.

Each system was established under aerobic and anaerobic conditions. To reach anaerobic conditions in an incubator, 100 μL of sterile mineral oil was added after inoculation. The perimeter of the plate was covered with a 3 mm deep band of smooth modeling clay, and vacuum grease was applied over the clay. In parallel, the hermetically sealable bag (GasPack, BD, Franklin Lakes, NJ, USA) was sterilized with UV for 20 min. In the sterile bag, a gas-generating cartridge (NIKI-MLT, Saint-Petersburg, Russia) was placed immediately before plate incubation. The plate without a cover lid was immediately placed in the prepared bag. Then, the cover lid of the plate was immediately placed into the bag to avoid contamination with grease and to leave the plate open for O_2_ elimination. The bag was closed tightly and left for 2 h at RT to eliminate O_2_. After incubation, the plate was tightly closed with the lid without disrupting of the bag seal, and the plate was removed from the bag and the perimeter of the lid was covered with grease.

The plates were incubated in a Xmark microplate spectrophotometer (Bio-Rad, Hercules, CA, USA) at 33.5 °C with shaking (model I) or without shaking (model II) for 72 h. Additionally, biofilms on the bottom of the wells were measured after incubation; for this measurement, the supernatant was removed from the wells, 200 μL of fresh sterile RCM was added to each well, and then, the absorbance was read at OD_540_ for each measurement.

The following kinetic parameters of the growth curve were analyzed: maximal growth rate (h^−1^), minimal generation time (h), and maximum OD_540_. OD of culture was proposed to be proportional to cell amount in suspension. Thus, the maximal growth rate was calculated by the formula µ = Ln((OD2/OD1))/(T2-T1)) on the linear portion of the semilogarithmic plot of the growth curve. The linear portions were calculated using the SLOPE function in MS Excel, calculates the coefficient of linear regression; thus, we identified the portion of the curve as “linear” when it has both a minimal standard deviation of the coefficient and when the coefficient is maximal. Doubling time (h) was calculated using the formula t = ln2/µ.

### 2.5. Biofilm Growth on Glass Microfiber Filters (GMFFs)

The GMFF method was originally developed by Plakunov et al. [19]. Briefly, bacterial suspensions were adjusted to OD_540_ = 0.5 and 1. Sterile GMFFs (21 mm in diameter, Whatman, Little Chalfont, UK) were placed in Petri dishes onto RCM agar with the addition of NE, and the controls were without the addition of the hormone. To obtain mono-species biofilms, 20 μL of suspension with OD_540_ = 0.5 was seeded onto a filter. To obtain dual-species biofilms, suspensions with OD_540_ = 1 were mixed together in equal proportions, and 20 μL of this mixture was inoculated. Incubation was performed under both aerobic and anaerobic conditions. Incubation was conducted at 33.5 °C for 24 and 72 h. For anaerobic conditions, a GasPack-Anaerogaz system (NIKI-MLT) was used. After incubation, a part of the filters with biomass was disrupted to count the colony forming units (CFU). A filter was placed into a tube with 10 mL of sterile PS and was disrupted with a sterile glass stick and then vortexed for 1 min to make a homogenous suspension. The absence of cell aggregates was monitored microscopically. Then, 20 µL of this prepared suspension was diluted and seeded in plates LB agar. The number of colonies was calculated after 2 days of incubation at 33.5 °C.

To estimate the metabolic activity of cells in biofilms, filters were stained with 3-(4,5-dimethyl-2-thiazolyl)-2.5-diphenyl-2*H*-tetrazolium bromide (MTT). MTT solution (0.1% in sterile LB medium) was prepared in advance and stored at −18 °C. As an electron acceptor MTT transforms to insoluble formazan and this reaction depends on the metabolic activity of the cells. Formazan was extracted in DMSO for 24 h, and the absorbance of the extract was measured at OD_590_.

### 2.6. Statistics and Data Processing

All experiments were conducted at least in triplicate. The nonparametric Mann-Whitney U test was performed for statistical data evaluation. All statistical and microbiological data plots were generated in GraphPad Prism 2007 software (GraphPad Software, La Jolla, CA, USA). Kinetic growth parameters were calculated in the Microsoft Excel 2007 software (Microsoft Corporation, Redmond, WA, USA). Where appropriate, average relative values (control without addition of epinephrine was designated as 100%) and absolute values were plotted on the graphs, and the standard error of the mean was depicted as error bars. Data were considered to be statistically significant at a confidence level of 95% (*p*-value < 0.05).

## 3. Results

### 3.1. PTFE Cubes Assay

First, to identify the potential effect of NE on both planktonic and biofilm growth at the same time we decided to test a physiological concentration (3.5 × 10^−9^ M) and three increased concentrations of NE in single cultures to model human stress conditions. The growth of planktonic cultures of *S. aureus* did not change significantly under aerobic conditions but were stimulated slightly at physiological concentrations of NE in anaerobic cultures (121.3 ± 14.2%) after 24 h of incubation (Figure 1A). At the same time, we revealed biofilm stimulation of this organism under aerobic conditions for 72 h. Biofilm growth was induced at the physiological level of NE (130.5 ± 25%), and then, the growth was reduced to the control level and was induced again with increasing concentrations of NE (Figure 1C). After 24 h, the physiological NE concentration alone had no significant effect on biofilm growth (111.3 ± 8.6%). The different effects of NE on *S. aureus* biofilms were revealed under anaerobic conditions. Biofilm stimulation was observed after 24 h of incubation, but the threshold concentration was higher (3.5 × 10^−8^ M) than physiological levels. However, after 72 h of incubation, biofilm growth was strongly inhibited, mostly at concentration of 3.5 × 10^−7^ M (49.9 ± 4.2%).

The opposite effect of NE has been shown on *S. epidermidis.* Biofilm growth was inhibited after 24 h under aerobic conditions (Figure 1B), especially at a concentration of 3.5 × 10^−7^ (77.7 ± 13.8%), whereas planktonic culture was not affected after the same incubation time. After 72 h, biofilms were inhibited in the same manner, but this inhibition was less significant and more gradual (Figure 1D). In comparison with *S. aureus,* NE slightly suppressed biofilm growth under anaerobic conditions in the same manner in *S. epidermidis*. Biofilm growth was inhibited at 3.5 × 10^−6^ M (70.9 ± 5.2%), whereas planktonic growth was slightly stimulated (110.5 ± 2.9% at 3.5 × 10^−6^ M and 107.6 ± 3.4% at 3.5 × 10^−7^ M NE).

Considering these results, we decided to use a concentration of 3.5 × 10^−7^ M for further studies because this concentration of NE had the most significant and variable effect on *S. aureus* biofilms (from strong inhibition to stimulation), which makes this more efficacious in studying dual-species biofilms. First, we decided to test whether the effect of NE is strongly related to biofilm formation or whether NE mostly plays a role as a nonspecific biofilm modulator. To make this determination, we performed a kinetic analysis of cell growth.

### 3.2. Study of the Kinetic Parameters of Planktonic Cultures and Biofilms

In this part of the work, we focused on studying the growth kinetic parameters of mono-species and dual-species staphylococcal cultures in the presence of selected NE concentrations (3.5 × 10^−7^ M). Since both planktonic and biofilm populations were present in each well, we used model I and model II in which each population predominated, and each experiment was conducted under aerobic and anaerobic conditions. In model I, the planktonic culture was dominant, and the biofilm fraction was dominant in model II. NE had no effect on the planktonic growth of *S. aureus* culture under aerobic conditions but slightly induced the biofilm growth rate (doubling time was 2.48 h vs. 2.61 h in the control, with linear portion of the curve was 3.5 h vs. 4.25). In model II under anaerobic conditions, the linear portion of the culture growth rate curve increased (4.75 h in comparison with 4.0 in the control), and the maximal OD_540_ also increased (OD_540_ = 1.15 vs. 1.01 in the control), whereas the doubling time was the same. These results suggested that NE can induce biofilm growth at the late exponential phase, and they are in accordance with the stimulation we observed on PTFE cubes after 24 h in anaerobic conditions, but they were opposite to the biofilm inhibition on cubes after 72 h. Since biofilm growth tends to decline after 16 h of incubation (Appendix A), we suggest that NE acted in the same manner on cubes but in different time spans depending on the surface type.

The growth rate of planktonic cultures and biofilms of *S. epidermidis* was higher under both aerobic and anaerobic conditions (Appendix A). In model II, the linear portion of the growth curve was much longer (5.25 h vs. 3.5 h in control) under aerobic conditions, but the maximum growth rate was higher (OD _540_ = 1.11 vs. 1.01 in control). However, under anaerobic conditions, NE induced the growth of planktonic cultures and slightly inhibited biofilm growth in the stationary phase (Appendix A), as we previously showed with PTFE cubes. Additionally, the amount of biofilm on the bottom of the well was strongly reduced under these conditions (OD_540_ = 0.76 ± 0.04 vs. OD_540_ = 0.9 ± 0.02 in the control, data not shown). Therefore, we suggest that NE can affect the transition between planktonic and biofilm phenotypes.

The growth curve of the mixed culture had the same characteristics as that of *S. epidermidis* in most cases (Appendix A). However, we observed growth inhibition in model I in the stationary phase under anaerobic conditions (Appendix A). This result can probably be explained by the *S. epidermidis* culture likely being more sensitive to NE in these conditions in general, which is why the growth curve of the mixed culture resembled the same curve of *S. aureus*. To observe how NE affects the metabolic activity and species composition in dual-species biofilms, we performed tests on GMFFs.

### 3.3. Effect of NE on Biofilms on GMFFs

In this model, biofilms were grown without planktonic culture, which made this model more representative of native biofilms. Growing bacteria on filters is suitable for both staining and CFU counts because the filter itself is an abrasive material. The МTT staining results, indicating metabolic activity, were not significantly different in the biofilms. In general, we observed a significantly lower CFU count in *S. epidermidis* biofilms than in *S. aureus* in mono-species biofilms (1.5 ± 0.5 × 10^8^ vs. 7.1 ± 1.5 × 10^8^ after 24 h and 1.7 ± 0.5 × 10^8^ vs. 1.4 ± 0.1 × 10^9^ after 72 h under aerobic conditions, 1.4 ± 0.4 × 10^8^ vs. 3.7 ± 1.2 × 10^8^ after 24 h and 1.7 ± 0.2 × 10^6^ vs. 2.3 ± 0.7 × 10^8^ after 72 h under anaerobic conditions) indicating that *S. aureus* is more invasive without competitive strain. However, the CFU count of *S. aureus* was strongly decreased in dual-species biofilms (Figure 2A,C,E) by an order more than that of the mono-species biofilms (1.1 ± 0.1 × 10^7^ vs. 7.1 ± 1.5 × 10^8^ in the mono-species biofilms after 24 h, 7.1 ± 1.4 × 10^7^ vs. 1.4 ± 0.1 × 10^9^ after 72 h under aerobic conditions, 3.6 ± 0.9 × 10^7^ vs. 3.7 ± 1.2 × 10^8^ in the mono-species biofilms after 24 h, 4.5 ± 1.4 × 10^6^ vs. 2.3 ± 0.7 × 10^8^ after 72 h under anaerobic conditions), suggesting that *S. epidermidis* had an advantage and strongly suppressed the competitor. NE significantly increased this competitive growth suppression of *S. aureus* (2.7 ± 0.6 × 10^7^ vs. 7.1 ± 1.4 × 10^7^ in control) after 72 h of incubation under aerobic conditions (Figure 2C), but no effect was observed after 24 h. At the same time, the CFU count of *S. epidermidis* did not change significantly in dual-species biofilms, which is also evidence of the dominance of this skin commensal organism. In dual-species biofilms, the CFU count of *S. aureus* (1.1 ± 1 × 10^8^ after 24 h and 7.1 ± 1.4 × 10^7^ after 72 h) decreased under aerobic conditions over time, and NE enhanced this effect (9.5 ± 1.5 × 10^8^ after 24 h and 2.7 ± 0.6 × 10^7^ after 72 h). After 72 h of incubation in anaerobic conditions CFU count of *S. epidermidis* was increased in dual-species biofilms but NE increased this stimulation only in mono-species biofilms (Figure 2G).

## 4. Discussion

Interactions between cutaneous bacteria in microbial communities and the impact of human hormones on bacterial biofilms have attracted increasing interest in the last decade. Despite the discovered role of NE in iron uptake through interaction with the host protein transferrin or lactoferrin [12], there was no evidence showing an impact of this compound on bacterial behavior in multi-species biofilms of cutaneous bacteria at concentrations similar to physiological levels in plasma (less than 50 µM). The competitive suppression of invasive *S. aureus* by cutaneous *S. epidermidis* has been observed in previous studies [3,20], but little is known about the impact of hormones on this process. In this study, we focused on the impact of the human hormone norepinephrine on mono-species and dual-species biofilms of two closely related competitive species, *S. aureus* and *S. epidermidis*, in different systems.

First, we revealed a different effect of NE on staphylococci with strong dependence on culture conditions. Norepinephrine did not have a significant impact on the growth of planktonic cultures of *S. aureus* but induced planktonic growth of *S. epidermidis* under anaerobic conditions in the late stationary phase, which we also confirmed by kinetic analysis. Interestingly, biofilms were inhibited in all models in the presence of planktonic culture (PTFE cubes and partially in 96-well plates), but there was no effect of *S. epidermidis* biofilms on glass filters. This result indicates that NE can probably induce the transition from a biofilm phenotype to a planktonic phenotype, especially under anaerobic conditions, but without any apparent growth inhibition. Another explanation is that NE facilitates the dispersion of mature biofilms. This finding is of interest with respect to a study [21] showing that NE can reduce the adhesion of *Streptococcus pneumonia* to lung epithelial cells and indicates that NE has a more complex mechanism of action. For *S. aureus*, an even more pronounced dependence on culture conditions was observed for the NE effect on biofilm growth. On PTFE cubes, biphasic stimulation under aerobic conditions and strong inhibition under anaerobic conditions were revealed after 72 h of incubation, whereas after 24 h, there was only a mild increase in biofilm formation. Whereas this biphasic character of biofilm stimulation under aerobic conditions was more significant after 72 h, we suggest that NE can mostly affect biofilm growth and maturation under aerobic conditions. However, under anaerobic conditions, NE strongly inhibited biofilm growth after 72 h but induced biofilm growth after 24 h, which can be a sign of matrix inhibition or induction of biofilm dispersion over time. Additionally, no inhibition of biofilm formation was observed at the selected NE concentration in kinetic models. These results allow us to suggest that NE mostly affects biofilm growth and maturation rather than the initial attachment steps of biofilm formation or that both biofilms are more sensitive to such agents in anaerobic conditions. The kinetic analysis of mixed dual-species culture showed that *S. epidermidis* was dominant under all conditions except anaerobic conditions in model I, where it can be suppressed in the stationary phase. Thus, the strong dependence of NE action on cultivation conditions may indicate that NE plays a more significant role in central metabolism and requires further study.

In the second part of the work, we focused on the role of NE in the interaction of two competitive species in mixed biofilms on glass filters. As expected, *S. aureus* growth was suppressed by *S. epidermidis* in dual-species biofilms, and this suppression was stronger when incubated under aerobic conditions. This effect can be explained by (1) the higher growth rate of *S. epidermidis*, which we observed in kinetic models, and by (2) the probable production of antimicrobial agents, for example, the protease Esp, which has been described in a recent study [22]. The lower CFU count in *S. epidermidis* biofilms grown on filters can potentially be explained by the second proposition. NE can facilitate this suppression in dual-species biofilms probably by affecting metabolism. This mechanism is indicated by the fact that NE slightly induced the maximal growth rate in kinetic model I but suppressed the CFU count in biofilms on filters after 72 h. It is reasonable to assume that NE can accelerate the metabolism of *S. epidermidis*, probably in the stationary phase under anaerobic conditions. Since the МТТ staining results did not corroborate this probable acceleration, further research is needed for a deeper understanding of the mechanism of NE action.

## 5. Conclusions

Considering the results obtained, we can conclude that the human stress hormone and neurotransmitter norepinephrine is involved in biofilm regulation of *S. aureus* and *S. epidermidis* with a strong dependence on culture conditions. *S. epidermidis* suppresses *S. aureus* growth in dual-species biofilms, and norepinephrine can accelerate this process.

## Figures and Tables

**Figure 1 microorganisms-09-00820-f001:**
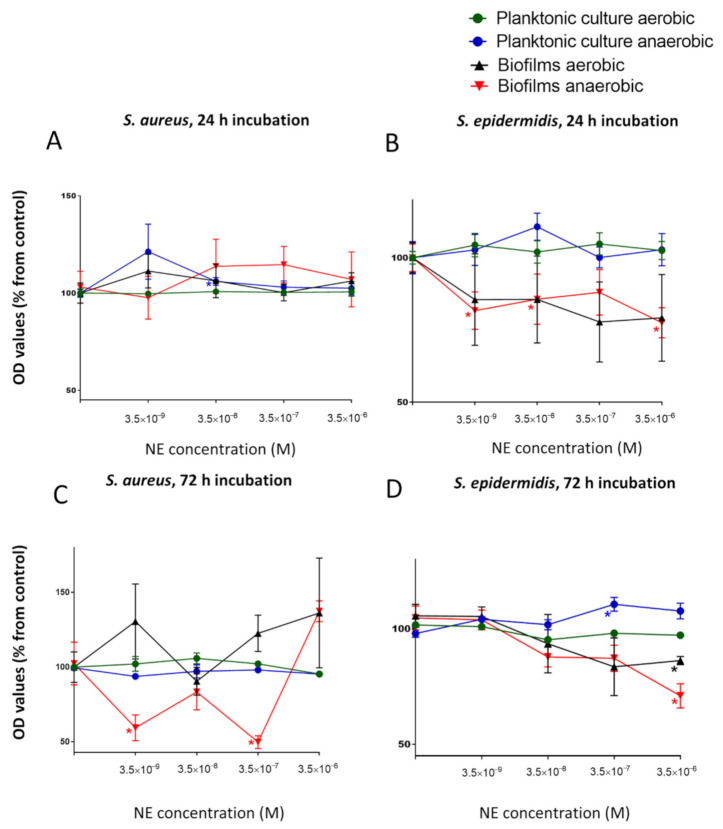
Impact of a physiological concentration (3.5 × 10^−9^ M) and three increased NE concentrations (3.5 × 10^−8^. 3.5 × 10^−7^, 3.5 × 10^−6^ M) on planktonic growth and biofilm formations of *S. aureus* (**A**,**C**) and *S. epidermidis* (**B**,**D**) after 24 h (**A**,**B**) and 72 h (**C**,**D**) of incubation. Error bars represent standard errors of the mean, *—*p* < 0.05 according to Mann-Whitney test.

**Figure 2 microorganisms-09-00820-f002:**
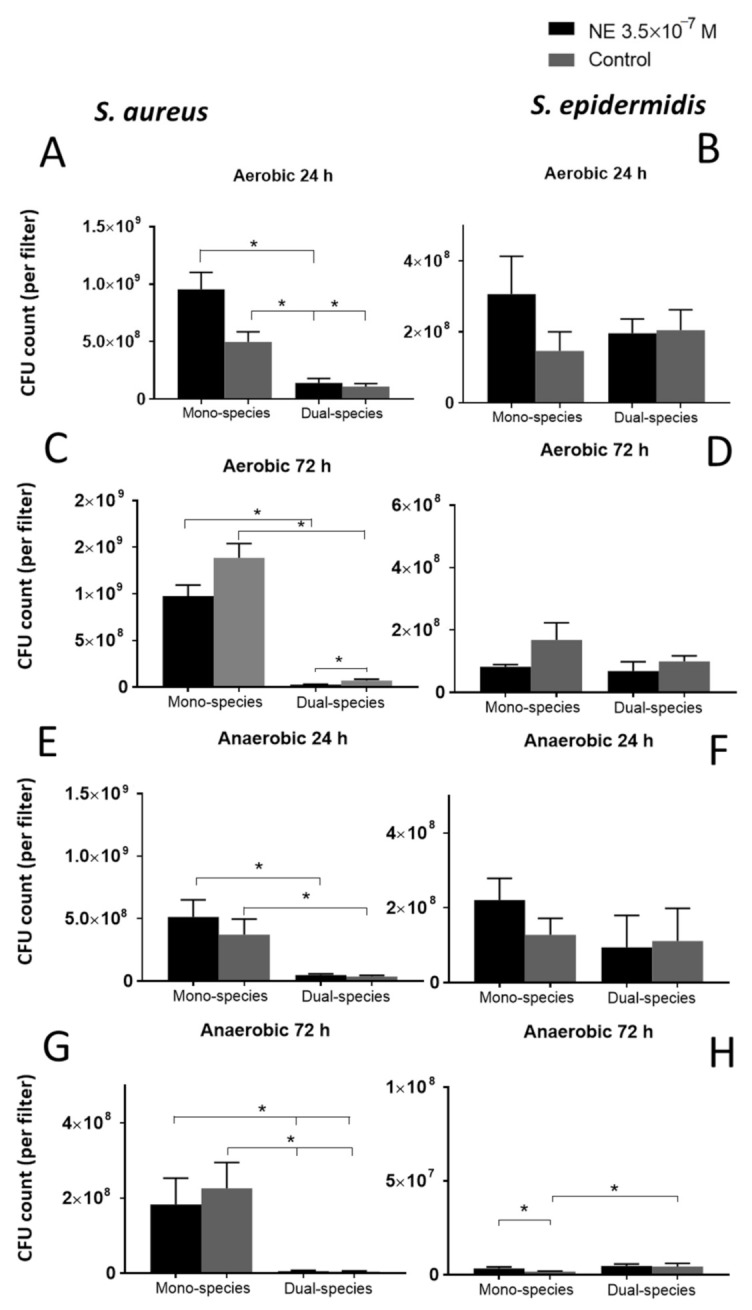
Total CFU count of *S. aureus* (**A**,**C**,**E**,**G**) and *S. epidermidis* (**B**,**D**,**F**,**H**) per filter in studied biofilms under aerobic (**A**–**D**) and anaerobic (**E**–**H**) conditions after 24 h (**A**,**B**,**E**,**F**) and 72 h (**C**,**D**,**G**,**H**) of incubation. Mean values are indicated, and error bars represent standard error of the mean, *—*p* < 0.05 according to Mann-Whitney test.

## Data Availability

Not applicable.

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
