# Peer review of "The Impact of Norepinephrine on Mono-Species and Dual-Species Staphylococcal Biofilms"

_microorganisms, 2021, doi:10.3390/microorganisms9040820_

Round 1
Reviewer 1 Report
The manuscript presented for research describes the effect of norepinephrine on mono-species and dual-species staphylococcal biofilms.
As a reviewer, I have a few questions for the authors concerning mainly research methodology.
Why was the S. epidermidis ATCC 14990 strain used in the study ?. The reference for biofilm formation is S. epidermidis ATCC 35984 or ATCC 35983 strain. How do you know that this strain forms a biofilm?
Why was RCM medium used ?, usually MHB or TSB medium is used for activity studies.
Why was the temperature used at 33.5 and not 37 degrees. According to the ATCC requirements for these reference strains, conditions should be aerobic with a growth temperature of 37 degrees. Line 65, 85, 124, 146, 153.
Why was the density measured at OD 540 and not as assumed at OD 600? Line 83,85,97,98,105,130, 140, 143.
Why was monospecies culture assumed to have a suspension density that was half that of dual-species? This significantly disturbs the obtained results.
For the results presented in Figure 1, correlation analyzes were not taken into account.
Author Response
Reviewer 1
The manuscript presented for research describes the effect of norepinephrine on mono-species and dual-species staphylococcal biofilms. As a reviewer, I have a few questions for the authors concerning mainly research methodology.
Dear Reviewer,
We thank you for valuable questions and сomments and we are hopeful they will make our manuscript better. Comments and our responses are listed below.
- Why was the epidermidis ATCC 14990 strain used in the study? The reference for biofilm formation is S. epidermidis ATCC 35984 or ATCC 35983 strain. How do you know that this strain forms a biofilm?
Thank you for your valuable question. This strain was isolated from nasal mucosa, whereas two other strains you mentioned are isolated from catheter surface, that is why strain ATCC 14990 is more suitable for our study. We agree that our choice was not evident and made an appropriate changes in manuscript (line 65, 66). This strain is able to form biofilms which has been shown in our experiments on PTFE cubes and in other model systems.
- Why was RCM medium used?, usually MHB or TSB medium is used for activity studies.
We thank you for this important notification. According to our previous studies, RCM is the most appropriate medium for cutaneous bacteria, especially Cutibacterium acnes. Since we study various cutaneous bacteria in multi-species biofilms under aerobic and anaerobic conditions, we use one standard medium for correct comparison of our results (Gannesen et al., 2018). We also added a link on our recent study in the reference list (Ovcharova et al., 2020, line 404) and added a link to our previous studies in the methodological section (line 73).
- Why was the temperature used at 33.5 and not 37 degrees. According to the ATCC requirements for these reference strains, conditions should be aerobic with a growth temperature of 37 degrees. Line 65, 85, 124, 146, 153.
The temperature of 33.5 degrees was used as it close to the skin’s physiological temperature as it was mentioned in the text (line 69). The same temperature was used in our previous studies (Gannesen et al., 2018). In this study we use both aerobic and anaerobic conditions to model various skin regions. We are very grateful for this comment and made an explanation in line 62.
- Why was the density measured at OD 540 and not as assumed at OD 600? Line 83,85,97,98,105,130, 140, 143.
We thank you for this important notification. This wave length (540 nm) is more suitable for optical density measurement because of the maximal light scattering value in colloid bacterial suspension. It also commonly used in different studies for years (Bögli‐Stuber et al., 2005; Paschoal et al., 2013; Depan, Mirsa, 2014; Larcher, Yargeau, 2017). We are informed that OD 600 is concerned to be a common widely used because of the minimal absorption of the medium itself. However, in our studies we used a sterile medium as a reference and therefore preferred to measure OD 540 as more precise value. An appropriate change was made in the methodological section (line 91).
- Why was monospecies culture assumed to have a suspension density that was half that of dual-species? This significantly disturbs the obtained results.
Thank you for the question. This procedure is needed to obtain an equal amount of cells in both mono-species and dual-species cultures. When mixed, the total suspension density of dual-species culture is in 2 times more, but density of each culture in the mixture becomes the same as in mono-species suspension.
6) For the results presented in Figure 1, correlation analyzes were not taken into account.
Thank you for this important comment. The results of Mann-Whitney test were indicated in Figure 1.
Additionally, we apologize for the miscalculations in «Results» section. Figure 2G, H has been changed. We kindly ask you to pay attention on these changes in the text (line 268, 273). Other data were checked again, and we were convinced of their reliability. Also we have corrected two misprints (line 257, 314) and made three additional corrections (line 81, 280).
Reviewer 2 Report
Manuscript by Mart'yanov et al describes the effect of norepinephrine on the growth of planktonic cultures and monospecies and dual-species biofilms of S. epidermidis and S. aureus strains. Overall, this is an interesting study. The introduction and the discussion are clear; however, the “Material and Methods” and “Results” sections are a bit confusing and I recommend a new reading and rephrasing of the text.
The abstract should be structured in order to have background, methods, results and conclusions.
Line 58: two strains individually and together to determine…
Line 104: The cultures were centrifuged for how long?
Line 125: Did you scrape the biofilm from the bottom of the well? Otherwise, I do not think that removing and adding new medium is enough for measuring biofilm formation.
Line 131: This sentence is confusing. Rephrase it.
Line 146: the incubation was done for how long? 24h?
Line 157: Rephrase the sentence
Line 171: The authors state that physiological concentration of NE and also three increased concentrations were used. However, none of this is mentioned in the Material and Methods section.
Figure 1 must be mentioned in the text.
Line 225: Something is missing in the sentence.
In the “Results” section the authors explain their ideas and conclusions. This information should be in the “Discussion” section that would also benefit from it since it is a bit short.
Line 285: the authors mention “previous studies” but cite only one study
Line 291: probably? Did NE induce planktonic growth of S. epidermidis or not?
Line 299: A study from 2013 is not a recent study.
The bibliography should be revised in the text and in the reference section since, for example, in line 299 the refence includes name and date instead of number.
Author Response
Dear reviewer,
Thank you for the important notices and remarks. We addressed the points mentioned and collected the answer.
Reviewer 2
Manuscript by Mart'yanov et al describes the effect of norepinephrine on the growth of planktonic cultures and monospecies and dual-species biofilms of S. epidermidis and S. aureus strains. Overall, this is an interesting study. The introduction and the discussion are clear; however, the “Material and Methods” and “Results” sections are a bit confusing and I recommend a new reading and rephrasing of the text.
Dear Reviewer,
We thank you for valuable questions and сomments and we are hopeful they will make our manuscript better. Comments and our responses are listed below.
1) The abstract should be structured in order to have background, methods, results and conclusions.
Thank you, we added some methodological description in abstract (line14). However, we found other part of abstract to be in accordance to your feedback.
Line 58: two strains individually and together to determine…
Thank you very much for this remark. We changed this sentence to make it clear (line 60)
Line 104: The cultures were centrifuged for how long?
Thank you for the question, cultures were centrifuged for 15 min. We changed the text accordingly (line 109)
Line 125: Did you scrape the biofilm from the bottom of the well? Otherwise, I do not think that removing and adding new medium is enough for measuring biofilm formation.
Thank you for your important question. Scraping the biofilm is strongly unnecessary for our cultures because of the high density of biofilm matrix. This would lead to large variations of experimental data. At the same time our cultures form a dense pellicle-like biofilms on the well’s bottom, that’s why we consider new medium adding to be more reliable for measuring biofilm formation in 96-well plates. Please kindly note, that this situation is strongly specific for our culture but not, for instance, for E. coli.
Line 131: This sentence is confusing. Rephrase it.
Thank you for this kind remark. The sentence has been rephrased (line 135).
Line 146: the incubation was done for how long? 24h?
Thank you for the question. Incubation was done for 24 h and 72 h (Figure 2). We have made a change in the text (line 155)
Line 157: Rephrase the sentence
Thank you, we rephrased the sentence (line 166)
Line 171: The authors state that physiological concentration of NE and also three increased concentrations were used. However, none of this is mentioned in the Material and Methods section.
Thank you, we changed the text (line 79)
Figure 1 must be mentioned in the text.
Thank you, we made a few changes in the text (line 186, 190, 205, 208)
Line 225: Something is missing in the sentence.
Thank you, we added a missing word (line 237)
In the “Results” section the authors explain their ideas and conclusions. This information should be in the “Discussion” section that would also benefit from it since it is a bit short.
Thank you, some sentences were removed from «Results» section (lines 195-203) and we have completed the «Discussion» (lines 317-322).
Line 285: the authors mention “previous studies” but cite only one study
Thank you, we added one more citation (line 297)
Line 291: probably? Did NE induce planktonic growth of S. epidermidis or not?
Thank you, we made a correction in the sentence (line 303)
Line 299: A study from 2013 is not a recent study.
Thank you, since we found this citation important, just «recent» was removed (line 311)
The bibliography should be revised in the text and in the reference section since, for example, in line 299 the reference includes name and date instead of number.
Thank you, we checked the manuscript and made only single correction (line 311)
Additionally, we apologize for the miscalculations in «Results» section. Figure 2G, H has been changed. We kindly ask you to pay attention on these changes in the text (line 268, 273). Other data were checked again, and we were convinced of their reliability. Also we have corrected two misprints (line 257, 314) and made three additional corrections (line 81, 280).
Best regards,
Andrei
Round 2
Reviewer 1 Report
Dear Authors.
Thank you very much for sending your reply.
On the use of a reference strain for research. The biological characteristics of the strain are important, not its primary isolation site. The S. epidermidis ATCC 14990 strain does not mention in the specification of the American Type Culture Collection that it is a reference strain for biofilm analysis, like the strains mentioned earlier in the review. In my opinion it cannot be used for this research.
I also disagree with the statement that RCM medium is better for assessing the growth of S. epidermidis and S. aureus strains. The authors themselves indicate that it is suitable for the growth of other microorganisms, such as Cutibacterium acnes. I believe that the media should be optimal for the tested microorganisms and the selection of the RCM medium is a wrong assumption.
In my opinion, it is also a mistake to double the number of cells for testing in a two-species biofilm. This results in a much faster consumption of nutrients, which may lead to erroneous results. I believe that bacterial innoculum of equal density should be used for analysis purposes.
I believe that this work should be subject to an independent methodological review.
Author Response
Dear Reviewer,
Thanks a lot for your important questions. Our answers and remarks are in the attached file.
Best regards,
Andrei

Round 3
Reviewer 1 Report
Dear authors,
Thank you for the satisfactory answers. I wish you satisfaction with your scientific work.